# THE CONJUGATE KERNEL FOR EFFICIENT TRAINING OF PHYSICS-INFORMED DEEP OPERATOR NETWORKS

**Amanda A. Howard**    **Saad Qadeer**    **Andrew W. Engel**    **Adam Tsou**

**Max Vargas**    **Tony Chiang**

**Panos Stinis**
Pacific Northwest National Laboratory
Richland, WA 99354, USA
`{amanda.howard, tony.chiang, panagiotis.stinis}@pnnl.gov`

## ABSTRACT

Recent work has shown that the empirical Neural Tangent Kernel (NTK) can significantly improve the training of physics-informed Deep Operator Networks (DeepONets). The NTK, however, is costly to calculate, greatly increasing the cost of training such systems. In this paper, we study the performance of the empirical Conjugate Kernel (CK) for physics-informed DeepONets, an efficient approximation to the NTK that has been observed to yield similar results. For physics-informed DeepONets, we show that the CK performance is comparable to the NTK, while significantly reducing the time complexity for training DeepONets with the NTK.

## 1   INTRODUCTION

One current research focus in scientific machine learning is physics-informed neural networks (PINNs, (Raissi et al., 2019)) and deep operator networks (DeepONets, (Lu et al., 2019)). While PINNs and physics-informed DeepONets (PI-DeepONets, (Wang et al., 2021; Goswami et al., 2022a)) have shown tremendous promise across a wide range of applications, e.g., Goswami et al. (2022b); Hao et al. (2023); Koric & Abueidda (2023); Karniadakis et al. (2021), training PI-DeepONets can be difficult and can result in large errors when compared with exact solutions (Wang et al., 2022b). Recent work has focused on efficient methods for improving accuracy of PINNs and PI-DeepONets, some of which include schemes that implement causality (Wang et al., 2022a), adaptive point selection methods (Wu et al., 2023), iterative methods to successively reduce the prediction in errors (Ainsworth & Dong, 2021; 2022; Howard et al., 2023; Aldirany et al., 2023; Wang & Lai, 2023), and adaptive weighting schemes (McClenny & Braga-Neto, 2023).

One method that has emerged for increasing training accuracy is the use of the Neural Tangent Kernel (NTK) as a weighting scheme in the loss function in PINNs (Wang et al., 2022c) and PI-DeepONets (Wang et al., 2022b). While the NTK significantly increases the accuracy, it greatly increases the computational cost by approximately a factor of six for each iteration. In this work, we present the use of the conjugate kernel (CK), an efficient approximation of the NTK, for adaptive weighting for training PI-DeepONets. While significantly less expensive to compute, we show that the CK provides almost equally accurate results for PI-DeepONets.

## 2   RELATED WORKS

### 2.1   CONJUGATE KERNEL AND THE NEURAL TANGENT KERNEL

The NTK (often called the empirical NTK) for a deep neural network (DNN) or a DeepONet is defined as the Gram matrix of the Jacobian of the DNN with respect to the network parameters (Jacot et al., 2018), originally defined for infinitely-wide neural networks (NNs). The NTK after

training a finite-width NN is referred to as the after kernel (Long, 2021). The CK (Fan & Wang, 2020; Wang et al., 2022d) is a "0-order" approximation to the NTK, defined as the Gram matrix of the parameters of the last layer of a FFNN (Hu & Huang, 2021). Here, we are only interested in *kernels after training*, and so both the NTK and CK will denote empirical kernels obtained from finite-width NNs after some amount of training.

Mathematically, the NTK is represented as the kernel induced by the map

$$\Phi_{\text{NTK},k} := \frac{\partial f_{\text{NN}}^{\theta}}{\partial \theta_k}, \quad k = 1, 2, \ldots, |\theta| \tag{1}$$

for a feedforward neural network $f_{\text{NN}}^{\theta}$ with trainable parameters $\theta$. The (empirical) NTK is then given by

$$\text{NTK}(\boldsymbol{z}, \widetilde{\boldsymbol{z}}) := \Phi_{\text{NTK}}(\boldsymbol{z})^{\top} \Phi_{\text{NTK}}(\widetilde{\boldsymbol{z}}) \tag{2}$$

Denoting the parameters in the last layer of $f_{\text{NN}}^{\theta}$ by $\theta^L$, the CK is the kernel induced by the map

$$\Phi_{\text{CK},k} := \frac{\partial f_{\text{NN}}^{\theta}}{\partial \theta_k^L}, \quad k = 1, 2, \ldots, |\theta^L|. \tag{3}$$

The CK is then defined as

$$\text{CK}(\boldsymbol{z}, \widetilde{\boldsymbol{z}}) := \Phi_{\text{CK}}(\boldsymbol{z})^{\top} \Phi_{\text{CK}}(\widetilde{\boldsymbol{z}}). \tag{4}$$

If we define E as the contribution to the NTK for all but the last layer, we can write the relationship between the NTK and CK as $\text{NTK}(\boldsymbol{z}, \widetilde{\boldsymbol{z}}) = \text{CK}(\boldsymbol{z}, \widetilde{\boldsymbol{z}}) + \text{E}(\boldsymbol{z}, \widetilde{\boldsymbol{z}})$.

The NTK has been shown to significantly improve the convergence of physics-informed training, e.g., Wang et al. (2022c;b); Howard et al. (2023); Bai et al. (2023), however, it is extremely computationally expensive to compute, thereby greatly increasing the training time of the NN or operator network. The CK has been shown to be an efficient approximation of the NTK, resulting in improved robustness, better conditioning, and increased accuracy in some test cases (Qadeer et al., 2023).

## 2.2 PHYSICS-INFORMED DEEP OPERATOR NETWORKS

In operator learning (Li et al., 2020a; Lu et al., 2021; Li et al., 2020b; Wen et al., 2022) a map is learned between two Banach spaces. For a general parametric PDE of the form $\mathcal{N}(u, s) = 0$ with boundary conditions $\mathcal{B}(u, s) = 0$ the operator network learns the PDE solution, denoted by $\mathcal{G}(u) = s(u)$ for $\mathcal{G} : \mathcal{U} \to \mathcal{S}$, where $\mathcal{U}$ is the space of input parameters, e.g., the initial conditions, with $u \in \mathcal{U}$, and $\mathcal{S}$ is the space of PDE solutions on a domain $\Omega$. $s \in \mathcal{S}$ is an unknown function governed by the PDE.

A standard DeepONet consists of the branch and trunk networks, which for our applications are both feedforward neural networks. The input to the branch network is the initial condition $u \in \mathcal{U}$, discretized at a set of $M$ discrete points, or sensor locations. The input to the trunk network is the spatial and time coordinates. The trunk and branch networks are trained simultaneously, and the DeepONet output is given by

$$\mathcal{G}_{\theta}(\mathbf{u})(\mathbf{x}) = \sum_{k=1}^{p} b_k(u_1, \ldots, u_M) t_k(\mathbf{x}) \tag{5}$$

where $\theta$ denotes the trainable parameters of the unstacked DeepONet, $b_k$ is the $k$-th component of the branch output and $t_k$ is the $k$-th component of the trunk output (Lu et al., 2019; 2021). In this work we use "modified" DeepONets (Wang et al., 2022b), an architecture that introduces encoder layers for the branch and trunk nets and has been shown to increase the accuracy of PI-DeepONet training. PI-DeepONets are trained *without* data to satisfy the given PDE through the use of automatic differentiation to calculate the derivatives. The loss function is given by (Wang et al., 2022b):

$$\mathcal{L}(\theta) = \frac{1}{N^*} \sum_{i=1}^{N^*} \lambda_i \left[ \mathcal{T}^{(i)} \left( \mathbf{u}^i, \mathcal{G}_{\theta}(\mathbf{u}^i)(\mathbf{x}^i) \right) \right]^2. \tag{6}$$

where $\lambda_i$ are weighting terms and where $N^* = NR$, $N$ is the number of samples considered as initial conditions, and $R$ is the number of collocation points at which the residual, initial conditions,

and boundary conditions are sampled. $\mathcal{T}^{(i)}$ denotes the operators in the loss function, including the boundary condition and differential operator. The weighting terms $\lambda_i$ in Eq. 6 require careful tuning to increase the training accuracy. Recent work has focused on the use of the NTK to successfully increase the accuracy for PI-DeepONets by locally choosing the optimal weights (Wang et al., 2022c; 2021). The NTK matrix is found as

$$H_{ij}^{NTK}(\theta) = \left\langle \frac{d\mathcal{T}^{(i)}(\mathbf{u}^i, \mathcal{G}_\theta(\mathbf{u}^i)(\mathbf{x}_i))}{d\theta}, \frac{d\mathcal{T}^{(j)}(\mathbf{u}^j, \mathcal{G}_\theta(\mathbf{u}^j)(\mathbf{x}_j))}{d\theta} \right\rangle. \tag{7}$$

Then, we can define the adaptive weights at a given iteration $n$ by

$$\lambda_i = \left( \frac{\max_{1 \le i \le N^*} H_{ii}(\theta_n)}{H_{ii}(\theta_n)} \right)^\alpha \tag{8}$$

where $H_{ii}$ is calculated using the NTK. The exponent $\alpha$ is generally chosen as $\alpha = 0.5$ or $\alpha = 1$, depending on the task. The DeepONet is trained using the ADAM optimizer in Jax Bradbury et al. (2018), with adaptive weights given by the NTK or CK used at every iteration.

## 3 RESULTS

The CK matrix is defined in an identical manner as the NTK, however, the derivatives are calculated only with respect to the final layer of the branch and trunk networks, denoted by $\theta^*$:

$$H_{ij}^{CK}(\theta) = \left\langle \frac{d\mathcal{T}^{(i)}(\mathbf{u}^i, \mathcal{G}_\theta(\mathbf{u}^i)(\mathbf{x}_i))}{d\theta^*}, \frac{d\mathcal{T}^{(j)}(\mathbf{u}^j, \mathcal{G}_\theta(\mathbf{u}^j)(\mathbf{x}_j))}{d\theta^*} \right\rangle. \tag{9}$$

We can now compute the adaptive weighting given by Eq. 8 by setting $H_{ii}$ as either the NTK or the CK. In each case, we train with $\alpha = 0.5$ or $\alpha = 1$, and select the best value of $\alpha$ for this study. To provide a comparison with previous literature Wang et al. (2022b), we contrast the NTK and CK weighting with the case of using fixed weighting, chosen as $\lambda_r = 1$ and $\lambda_{bc} = 10$.

### 3.1 BURGERS EQUATION

The viscous one-dimensional Burgers equation given in Appendix A.2 with viscosity $\nu$ and periodic boundary conditions presents a challenge for DeepONets, as shown in Wang et al. (2022b; 2021). To provide a direct comparison, we follow the procedure as outlined in Wang et al. (2022b) to generate a training and testing set by selecting initial conditions $u(x)$ from a Gaussian random field, and train using the same network hyperparameters. Results comparing the CK, the NTK, and using predetermined fixed weights are shown in Table 1. The use of the CK not only leads to comparable or improved test accuracies compared to the NTK but requires only 36.8% of the training time.

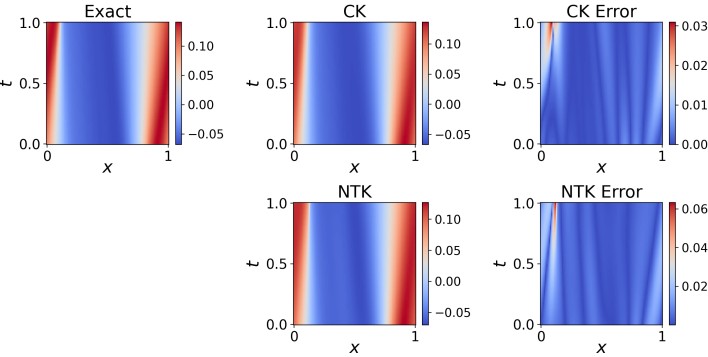

Figure 1: Sample results for viscous Burgers equation for $\nu = 0.0001$.

Table 1: Results for Burgers equation for a range of viscosities $\nu$. The mean and SD of the relative $\ell_2$ error are over 500 initial conditions in the test set. Mean run times are computed using an NVIDIA A100 GPU. The weighting scheme refers to the expressions chosen for $\lambda_i$ in Eq. 6.

| $\nu$ | Fixed weighting | NTK weighting | CK weighting |
|---|---|---|---|
| 0.01 | $3.65\% \pm 6.26\%$ | $0.63\% \pm 0.50\%(\alpha = 1)$ | $0.66\% \pm 0.57\%(\alpha = 1)$ |
| 0.001 | $9.18\% \pm 8.03\%$ | $2.80\% \pm 3.00\%(\alpha = 0.5)$ | $3.04\% \pm 3.82\%(\alpha = 0.5)$ |
| 0.0001 | $25.10\% \pm 9.53\%$ | $11.44\% \pm 5.91\%(\alpha = 0.5)$ | $8.34\% \pm 3.62\%(\alpha = 1)$ |
| Mean time (h) | 1.582 | 6.096 | 2.243 |

Table 2: Comparison of the relative $\ell_2$ error and computational time for the wave equation. The mean and SD of the relative $\ell_2$ error are over 500 initial conditions in the test set. Run times are computed using an NVIDIA A100 GPU. The weighting scheme refers to the expressions chosen for $\lambda_i$ in Eq. 6.

| | Fixed weighting | NTK weighting | CK weighting |
|---|---|---|---|
| Error | $3.45\% \pm 1.83\%$ | $0.93\% \pm 0.50\% (\alpha = 0.5)$ | $0.96\% \pm 0.74\% (\alpha = 1)$ |
| Run time (h) | 1.305 | 5.949 | 1.881 |

## 3.2 WAVE EQUATION

The wave equation is given by

$$s_{tt} = 2s_{xx}, \quad (x,t) \in \Omega = [0,1] \times [0,1] \tag{10}$$
$$s(x,0) = u(x), \quad x \in [0,1] \tag{11}$$
$$s_t(x,0) = 0, \quad x \in [0,1] \tag{12}$$
$$s(0,t) = s(1,t) = 0, \quad t \in [0,1]. \tag{13}$$

We take $u(x) = \sum_{n=1}^{5} b_n \sin(n\pi x)$. Then, the exact solution is given by $s(x,t) = \sum_{n=1}^{5} b_n \sin(n\pi x) \cos(n\pi\sqrt{2}t)$. To generate each initial condition $u$ we generate a set $\{b_n\}_{n=1}^{5}$ of normally distributed random variables. We train with 1000 random initial conditions as the training set.

In Table 2 we report the relative $\ell_2$ errors between the exact solution and results. We note that the CK error is very close to the NTK error, however, it takes only about 32% of the computational time to achieve results of this accuracy. While the CK does not offer additional accuracy over the NTK, the lower computational time is a compelling argument for why one may consider using it.

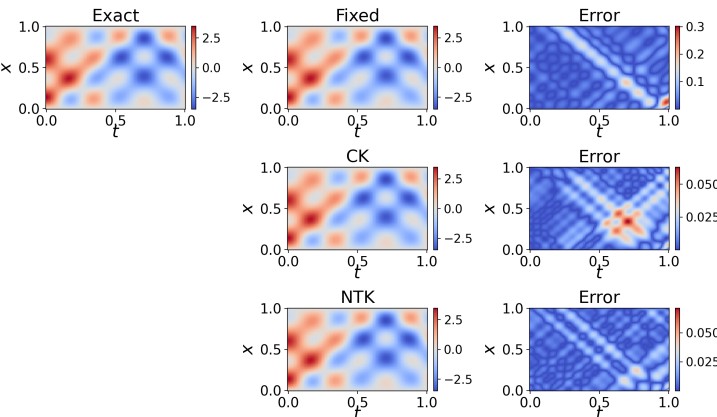

Figure 2: Sample results for the wave equation.

## 4 DISCUSSION

We have shown that the CK is an efficient approximation of the NTK, and can result in similar training accuracy for physics-informed DeepONets, while significantly reducing the computational time. In the cases presented, the CK results have errors quite similar to the NTK results, including outperforming the NTK for the smallest viscosity for the viscous Burgers equation. While the accuracy is similar between the two, using the CK rather than the NTK cuts the training time by roughly two-thirds, and for this reason, the CK should be considered as an alternative for the NTK when training physics-informed DeepONets. Testing the CK for two cases limits the generalization in this study, however the use of the CK will be further explored in future work, including tuning for the optimal value of the exponent $\alpha$ to ensure the robustness of results.

### ACKNOWLEDGMENTS

The work of SQ is supported by the Department of Energy (DOE) Office of Advanced Scientific Computing Research (ASCR) through the Pacific Northwest National Laboratory Distinguished Computational Mathematics Fellowship (Project No. 71268). The work of AD, MV, AE, and TC were partially supported by the Mathematics for Artificial Reasoning in Science (MARS) initiative via the Laboratory Directed Research and Development (LDRD) Program at PNNL. The work of AH and PS is partially supported by the U.S. Department of Energy, Office of Science, Advanced Scientific Computing Research program under the Scalable, Efficient and Accelerated Causal Reasoning Operators, Graphs and Spikes for Earth and Embedded Systems (SEA-CROGS) project (Project No. 80278). The computational work was partially performed using PNNL Institutional Computing at Pacific Northwest National Laboratory. Pacific Northwest National Laboratory is a multi-program national laboratory operated for the U.S. Department of Energy by Battelle Memorial Institute under Contract No. DE-AC05-76RL01830.

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

## A  APPENDIX

### A.1  $\ell_2$ ERRORS

The errors reported are the relative $\ell_2$ errors, given by $\frac{||s(u)(\mathbf{x}) - \mathcal{G}_\theta(\mathbf{u})(\mathbf{x})||_2}{||s(u)(\mathbf{x})||_2}$.

### A.2  BURGERS EQUATION

Burgers equation as considered in Sec. 3.1 is given by

$$\frac{\partial s}{\partial t} + s\frac{\partial s}{\partial x} - \nu\frac{\partial^2 s}{\partial x^2} = 0, \ (x,t) \in [0,1] \times [0,1] \tag{14}$$

$$s(x,0) = u(x), \ x \in [0,1], \tag{15}$$

$$s(0,t) = s(1,t), \ t \in [0,1], \tag{16}$$

$$\frac{\partial s}{\partial x}(0,t) = \frac{\partial s}{\partial x}(1,t), \ t \in [0,1], \tag{17}$$

where $\nu$ is the viscosity.

### A.3  TRAINING PARAMETERS

| Parameter | |
|---|---|
| Learning rate | $(10^{-3}, 5000, 0.9)$ |
| Branch network size | $[101, 100, 100, 100, 100, 100, 100, 100]$ |
| Trunk network size | $[2, 100, 100, 100, 100, 100, 100, 100]$ |
| Activation function | `tanh` |
| Batch size | 10000 |
| Iterations | 200,000 |
| $\lambda_r$ | 1.0 |
| $\lambda_{bc}$ | 10.0 |
| $M$ | 101 |
| $R_{bc}$ | 100 |
| $R_{res}$ | 2500 |
| $N$ | 1000 |

Table 3: Training parameters for the DeepONet results in Sec. 3. For the learning rate, the triplet $(a, b, c)$ denotes the `exponential_decay` function in Jax with learning rate $a$, decay steps $b$, and decay rate $c$. $N$ is the number of initial conditions used in the training set. $M$ is the number of sensor locations at which the initial condition is evaluated. $R_{bc}$ and $R_{res}$ are the number of points at which the boundary conditions and residual are evaluated, respectively.

