# OpenReview forum: "The conjugate kernel for efficient training of physics-informed deep operator networks"
_ICLR.cc/2024/Workshop/AI4DiffEqtnsInSci — AI4DiffEqtnsInSci @ ICLR 2024 Poster_

### Official Review · Reviewer_aXup · 2024-02-15
**uses conjugate kernel (CK) for updating the loss term coefficients in physics-informed deeponets**

**Rating:** 4
**Confidence:** 5

**Review:**

This works uses conjugate kernel (CK) as an efficient approximation of neural tangent kernel or NTK when the latter is used to increase the accuracy of physics-informed deeopnets in operator learning. Using two examples on Burgers’ and wave equations, the authors show that using CK, while insignificantly affecting the accuracy of the model, can reduce the training times (compared to cases when NTK is used).

The main idea of the paper is to use the Gram matrix of the Jacobian of the DNN with respect to its last layers’ parameters. This matrix is used to tune the coefficients of the different terms that appear in the loss function (data loss terms and PDE residuals) of a PI-DeepONet. The contribution of this paper is to use eq. 8 instead of eq. 7 for updating the weights.

I do not believe the paper makes sufficient contributions to the field to warrant its presentation in this workshop. This is especially the case since the results section lacks rigor. It is unclear if this method can be scaled up to cases where, e.g., the PDE has 3D spatial variables or has multiple outputs (e.g., the navier-stokes equations). It is also unclear why the authors are comparing their approach to a fixed-weight, i.e., to be fair, you can compare your approach to an adaptive weight scheme that is not based on NTK or CK (e.g., based on the moving average of the gradient magnitudes which are already available during training).

---

### Official Review · Reviewer_ybPM · 2024-02-17

**Rating:** 8
**Confidence:** 4

**Review:**

**Summary of the paper:** The authors present a neural tangent kernel (NTK) approximation for training physics-informed deepOnet more accurately and faster, by computing a conjugate NTK with only the last layer of the network. The proposed method achieves similar performance as full NTK computations but allows for much faster training.

**Strength of the paper:** The paper does a good job at introducing the main differences between NTK and conjugate kernel, and the motivation behind developing fast NTK approximations. It also describes well key aspects of physics-informed deepOnet, and how they fit within NTK computations. The results are sound: the conjugate gradient achieves similar accuracy as NTK, with a significant training time gain.

**Weakness of the paper:** My understanding is that NTK/CK are only used to compute weighing terms in the loss function, but the deepOnet weights are updated using a standard gradient descent. For someone not familiar with NTK, this is somewhat confusing, as equation 1 and 3 may intuitively lead to think that NTK updates neural network weights in a different fashion than standard gradient descent. In table 1, it is also unfortunate to use the term "weight" to refer to $\lambda_i$, as this may be mistaken with the network weights/parameters.

I wish the authors had expanded on the accuracy results for CK and low viscosity Burgers equation. It is surprising that CK achieves better performance than NTK, given that CK is an approximation of NTK. Do the authors have an idea why this happens?

---

### Meta-Review · Area_Chair_h1vP · 2024-03-02

**Recommendation:** Accept (Poster)

**Metareview:**

The authors present a neural tangent kernel (NTK) approximation for training physics-informed deepOnet more accurately and faster, by computing a conjugate NTK with only the last layer of the network. The proposed method achieves similar performance as full NTK computations but allows for much faster training. Whilst the work is promising it is lacking in rigour in that the results are not put in context with a broader range of other approaches and it is not clear how well this method can scale. Nevertheless it is a good contribution to the community and should be accepted for the poster session.

---

### Decision · Program_Chairs · 2024-03-02

Accept (Poster)